# DNA-DetectLLM: Unveiling AI-Generated Text via a DNA-Inspired Mutation-Repair Paradigm

**Xiaowei Zhu[1,2], Yubing Ren[1,2]***, **Fang Fang[1,2],**
**Qingfeng Tan[3,4], Shi Wang[5], Yanan Cao[1,2]**
[1]Institute of Information Engineering, Chinese Academy of Sciences, Beijing, China
[2]School of Cyber Security, University of Chinese Academy of Sciences, Beijing, China
[3]University International College, Macau University of Science and Technology, Macau
[4] Cyberspace Institute of Advanced Technology, Guangzhou University, Guangzhou, China
[5] Institute of Computing Science, Chinese Academy of Sciences, Beijing, China
`{zhuxiaowei, renyubing}@iie.ac.cn`

## Abstract

The rapid advancement of large language models (LLMs) has blurred the line between AI-generated and human-written text. This progress brings societal risks such as misinformation, authorship ambiguity, and intellectual property concerns, highlighting the urgent need for reliable AI-generated text detection methods. However, recent advances in generative language modeling have resulted in significant overlap between the feature distributions of human-written and AI-generated text, blurring classification boundaries and making accurate detection increasingly challenging. To address the above challenges, we propose a DNA-inspired perspective, leveraging a repair-based process to directly and interpretably capture the intrinsic differences between human-written and AI-generated text. Building on this perspective, we introduce DNA-DetectLLM, a zero-shot detection method for distinguishing AI-generated and human-written text. The method constructs an ideal AI-generated sequence for each input, iteratively repairs non-optimal tokens, and quantifies the cumulative repair effort as an interpretable detection signal. Empirical evaluations demonstrate that our method achieves state-of-the-art detection performance and exhibits strong robustness against various adversarial attacks and input lengths. Specifically, DNA-DetectLLM achieves relative improvements of **5.55%** in AUROC and **2.08%** in F1 score across multiple public benchmark datasets. Code and data are available at `https://github.com/Xiaoweizhu57/DNA-DetectLLM`.

## 1 Introduction

The rapid advancement of large language models (LLMs) has created increasingly human-like textual content, substantially narrowing the distinguishable gap between AI-generated and human-written text. While these improvements have catalyzed significant technological breakthroughs, they simultaneously pose critical societal challenges, including misinformation dissemination, authorship ambiguity, and threats to intellectual property rights [2, 1, 13]. Consequently, there is an urgent and growing need for effective and reliable methods to accurately detect AI-generated text.

While significant research efforts have been dedicated to AI-generated text detection, existing methodologies typically adopt either training-based or training-free methods. Training-based methods [28, 16, 12, 39, 11] depend upon large volumes of annotated data, limiting their scalability and

---

*Corresponding author.

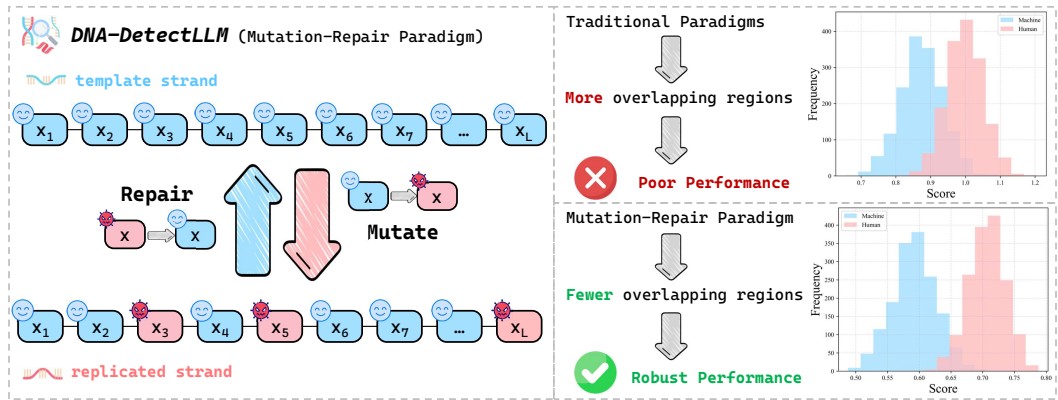

Figure 1: Illustration of the Mutation-Repair Paradigm. The input sequence can be analogized to a replicating strand, while the ideal AI-generated sequence corresponds to the template strand.

generalization to new domains. In contrast, training-free approaches [22, 4, 14, 36] leverage intrinsic statistical differences to distinguish human-written and AI-generated texts. Both paradigms fundamentally operate by attempting to identify distinct, separable boundaries within the feature space. However, recent advancements in generative language modeling have produced outputs increasingly indistinguishable from human-authored content, causing these classification boundaries to become progressively blurred. Empirical studies [7, 27] have highlighted substantial overlap regions in the feature distributions of human-written and AI-generated texts, significantly undermining detection accuracy in practical scenarios. Therefore, one capable of more precisely and intrinsically capturing differences between the generative processes of AI and human writing is urgently needed.

In molecular biology, DNA's double-helix structure ensures stable transmission of genetic information, yet mutations during replication introduce variations that can lead to individual differences or even diseases such as cancer. In a similar vein, an ideal AI-generated text sequence can be seen as a "template strand", representing the most probable token choices at each position. Human-written texts, by contrast, resemble mutated strands, where token selections deviate from the optimal probabilities, creating measurable differences. Inspired by this biological mechanism, we propose a new perspective for AI-generated text detection: by analogizing to DNA base-repair processes, we iteratively "correct" non-optimal tokens in a text and measure the difficulty of restoring it to the ideal AI-generated form. This repair-based approach captures the intrinsic divergence between AI-generated and human-written texts in a direct and interpretable manner.

Building on this intuition, we propose DNA-DetectLLM, a novel method for zero-shot detection of AI-generated texts. For each input sequence, we first construct its corresponding ideal AI sequence—that is, the sequence formed by greedily selecting the most probable token at each position under a reference language model. We then perform a token-by-token repair process on the input sequence, progressively modifying tokens toward their optimal choices until the sequence fully aligns with the ideal AI sequence. To quantify the difficulty of this repair process, we introduce a repair score that captures the cumulative effort required to complete the transformation. Finally, by comparing the repair score against a calibrated threshold, DNA-DetectLLM robustly distinguishes AI-generated texts from human-written ones, leveraging the fundamental differences in their deviation patterns from ideal generation.

DNA-DetectLLM consistently achieves state-of-the-art performance across multiple datasets and LLMs. In particular, it obtains relative improvements of **5.55%** in AUROC and **2.08%** in F1 score on three public benchmark datasets. Additionally, the method exhibits notable robustness against various adversarial attacks and across different input lengths. Efficiency experiments further indicate rapid detection capability, processing each sample in under 0.8s.

Our contributions are summarized as follows:

- Inspired by the mutation and repair mechanisms of nucleotide bases in DNA replication, we introduce the mutation-repair paradigm into AI-generated text detection.

- We propose DNA-DetectLLM, a novel zero-shot method for detecting AI-generated text that incrementally repairs mutated tokens within the input sequence until it perfectly aligns with the ideal AI-generated sequence, subsequently quantifying the repair difficulty as a metric for text detection.

- Extensive evaluations validate that DNA-DetectLLM offers a reliable, efficient, and broadly generalizable solution for AI-generated text detection, with consistent gains across various detection settings.

## 2  Related Works

Detecting AI-generated text is essential for enhancing public trust and preventing misuse, driving growing interest from both academia and industry. Beyond watermarking techniques [20], which embed identifiable markers during generation, current post hoc detection methods are broadly categorized into training-based and training-free methods.

**Training-based Methods.**   Such approaches typically involve training classification models to distinguish between AI-generated and human-written texts. Specifically, early efforts by OpenAI [28] employed RoBERTa-based models for training text classifiers. Subsequently, RADAR [16] introduced adversarial learning to enhance the robustness against paraphrased texts. DeTeCtive [12] utilized multi-level contrastive learning to map texts generated by different LLMs into corresponding feature spaces, classifying them based on similarity metrics. DPIC [39] extracted deep textual features by reconstructing prompts and regenerating texts. Biscope [11] proposed employing a bidirectional cross-entropy loss to extract statistical features for binary classifier training. R-Detect [29] employs a nonparametric kernel relative test to detect AI-generated text, thereby reducing the false positive rate compared to two-sample tests. However, existing research [5, 32] indicates that training-based methods consistently overfit to in-distribution features, resulting in poor generalization to out-of-distribution (OOD) texts. Consequently, researchers have increasingly focused on developing more universally applicable training-free methods.

**Training-free Methods.**   These training-free methods emphasize exploiting probabilistic characteristics of texts, constructing statistical scores based on specific hypotheses, and making decisions according to the comparison of scores against thresholds. For example, LogRank [9], Likelihood [15], and Entropy [17] calculate the average probability ranking, likelihood probabilities, and entropy values to measure the uncertainty of AI-generated texts. DetectGPT [22] pioneered a paradigm that uses perturbations to generate numerous contrast samples to evaluate the overall distribution. Although methods such as DetectLLM-NPR [30] and DNA-GPT [37] have further developed this paradigm, their efficiency limitations prevent real-time or large-scale detection implementations. Fast-DetectGPT [4] has since updated sampling techniques to compute conditional probability curvature, significantly improving detection efficiency and broadening potential applications. Binoculars [14] achieved state-of-the-art classification performance by calculating cross-perplexity from dual-model perspectives. Lastde++ [36] proposed focusing on local textual features by calculating Diversity Entropy to optimize classification performance.

## 3  DNA-DetectLLM

### 3.1  Preliminary

This study primarily involves two statistical metrics: log-perplexity, which quantifies the average token-level negative log-likelihood under a single model, and cross-perplexity, which captures the average per-token cross-entropy between the probability distributions of two models:

$$\log \mathrm{PPL}_{M_1}(s) = -\frac{1}{L} \sum_{i=1}^{L} \log P_{M_1}(x_i | x_{<i}),$$

$$\log \mathrm{X\text{-}PPL}_{M_1, M_2}(s) = -\frac{1}{L} \sum_{i=1}^{L} P_{M_1}(x_i | x_{<i}) \log P_{M_2}(x_i | x_{<i}),$$

(1)

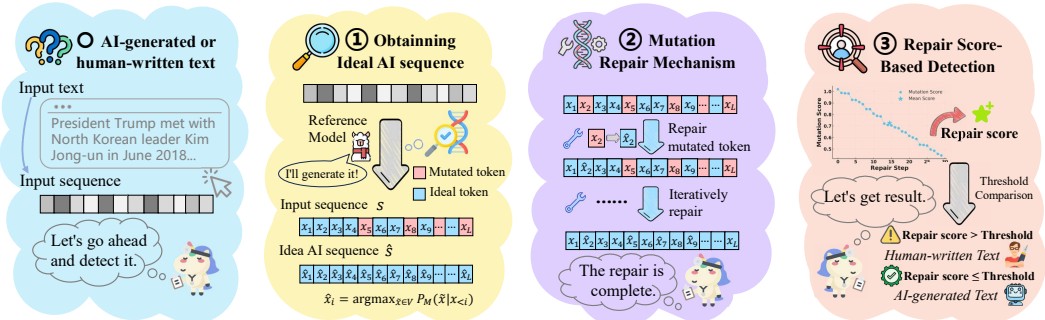

Figure 2: Overview of DNA-DetectLLM.

where $s$ is the input sequence of length $L$, $x_i$ denotes the $i$-th token, and $P_M(x_i|x_{<i})$ is the conditional probability of $x_i$ given its preceding tokens under reference model $M_1$ or observer model $M_2$. Furthermore, their ratio $\sigma(s)$ has been empirically demonstrated to serve as an effective score for distinguishing AI-generated text [14]. To quantify the effect of local token-level modifications on these metrics, this work introduces the **conditional log-perplexity** and **conditional score**:

$$\log \mathrm{PPL}_{M_1}(\tilde{s}|s) = -\frac{1}{L}\sum_{i=1}^{L} \log P_{M_1}(\tilde{x}_i|x_{<i}), \quad \sigma(\tilde{s}|s) = \frac{\log \mathrm{PPL}_{M_1}(\tilde{s}|s)}{\log \mathrm{X\text{-}PPL}_{M_1,M_2}(s)}, \tag{2}$$

where $\tilde{s}$ denotes the sequence obtained by modifying tokens in the input sequence $s$.

### 3.2 Overview of DNA-DetectLLM

The entire workflow of DNA-DetectLLM can be summarized in 3 key steps, shown in Figure 2.

**Step 1: Obtaining the ideal AI-generated Sequence.** We construct the ideal AI-generated sequence for a given input by greedily selecting the most probable token at each position.

**Step 2: Mutation Repair Mechanism.** We perform iterative token-level modifications on the input sequence until it fully aligns with the ideal AI-generated sequence.

**Step 3: Repair Score-Based Detection.** We introduce a repair score to quantify the difficulty of the repair, which is compared against a calibrated threshold to determine the detection result.

### 3.3 Obtaining the ideal AI-generated Sequence

We propose the concept of an *ideal AI-generated sequence* $\hat{s}$, analogous to the error-free template strand in DNA replication, where each token is selected by maximizing the conditional probability at its position:

$$\hat{s} = \{\hat{x}_1, \hat{x}_2, \ldots, \hat{x}_L\}, \quad \text{where } \hat{x}_i = \arg\max_{\tilde{x}\in\mathcal{V}} P_{M_1}(\tilde{x}|x_{<i}), \tag{3}$$

with $\mathcal{V}$ denoting the vocabulary and $x_{<i} = \{x_1, \ldots, x_{i-1}\}$ representing the preceding $i-1$ tokens of the input sequence $s$.

### 3.4 Mutation Repair Mechanism

We treat non-max probability tokens as mutated tokens and max-probability tokens as ideal tokens. Analogous to the mutation-repair paradigm in DNA, we propose a *mutation repair mechanism* aiming to uncover fundamental differences in deviation patterns between AI-generated and human-written texts. Under this mechanism, mutated tokens in the input sequence are iteratively repaired with their ideal tokens step by step, until the input fully aligns with the ideal form:

$$x_i \in s = \{x_1, x_2, \ldots, x_L\} \rightarrow \hat{x}_i = \arg\max_{\tilde{x}\in\mathcal{V}} P_{M_1}(\tilde{x}|x_{<i}), \quad \text{if } x_i \neq \hat{x}_i, \tag{4}$$

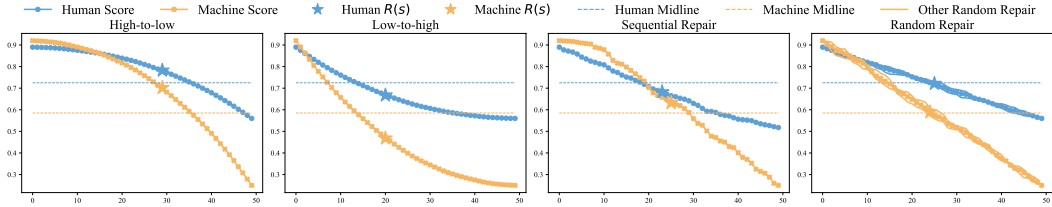

Figure 3: Variation of repair scores Across Different Repair Strategies.

## 3.5 Repair Score-Based Detection

We introduce the *repair score* $R(s)$ to quantify the difficulty of the repair process, defined as the average conditional score accumulated throughout the repair trajectory:

$$R(s) = \frac{1}{T+1} \sum_{t=0}^{T} \sigma(s_t|s) = \frac{\sum_{t=0}^{T} \log \text{PPL}_{M_1}(s_t|s)}{(T+1) \log \text{X-PPL}_{M_1,M_2}(s)}, \tag{5}$$

where $s_t$ is the sequence after $t$ repair steps, and $T$ is the total number of mutated tokens to be corrected.

Human-written texts typically exhibit more substantial mutations, resulting in greater repair difficulty. In contrast, AI-generated texts are generally easier to repair. Accordingly, the detection result for the input sequence is determined as:

$$\mathcal{D}(s) = \begin{cases} \text{Human-written Text}, & R(s) > \tau \\ \text{AI-generated Text}, & R(s) \leq \tau. \end{cases} \tag{6}$$

## 3.6 Sensitivity to Repair Order and Score Simplification

Figure 3 shows that different repair orders yield varying repair scores for the same input sequence, due to the unequal impact of each mutated token on the conditional score. Mutated tokens can be broadly categorized into high- and low-probability types. Repairing low-probability tokens typically causes larger shifts in the conditional score, while high-probability tokens lead to smaller changes. To systematically analyze the influence of repair order, we identify four types of principal repair strategies as follows:

- **High-to-low:** Repairing high-probability tokens first, followed by low-probability ones, results in a convex "repair curve" with a higher repair score.

- **Low-to-high:** Repairing low-probability tokens before high-probability ones yields a concave "repair curve" with a lower repair score.

- **Sequential Repair:** Tokens are repaired in their original order of appearance in the input sequence, regardless of their probability values.

- **Random Repair:** Tokens are repaired in a randomly chosen order. Averaging the repair scores across multiple random repairs leads to a more stable estimate.

These findings highlight the sensitivity of the repair score to the chosen repair order. Performing multiple random repairs effectively mitigates biases, resulting in a repair score close to the midpoint between the initial and final scores. Consequently, we further derive that the average repair score converges as the number of random repairs $N$ approaches infinity. The derivation is as follows:

Let $\{\delta_1, \delta_2, \ldots, \delta_T\}$ be a set of non-negative real numbers satisfying $\sum_{i=1}^{T} \delta_i = \sigma(s) - \sigma(\hat{s}|s)$. Moreover, $\delta_t = \sigma(s_{t-1}|s) - \sigma(s_t|s)$ quantifies the impact of repairing the current token on the score. Since the influence of each token repair on the conditional log-perplexity is independent and fixed, the set $\{\delta_1, \delta_2, \ldots, \delta_T\}$ consists of fixed values for a given input sequence, whose order varies depending on the repair strategy. It further follows:

$$\sigma(s_t|s) = \sigma(s) - \sum_{i=1}^{t} \delta_i \quad \text{for } t = 1, 2, \ldots, T, \quad \text{where } \sigma(s) = \sigma(s_0|s). \tag{7}$$

For a specific permutation $\phi \in [1, N]$, the corresponding repair score $R^\phi(s)$ is:

$$R^\phi(s) = \frac{1}{T+1} \sum_{t=0}^{T} \left( \sigma(s) - \sum_{i=1}^{t} \delta_i^\phi \right) = \sigma(s) - \frac{1}{T+1} \sum_{i=1}^{T} \delta_i^\phi \cdot (T - i + 1). \quad (8)$$

Since each token is equally likely to be repaired at random, the expected value of $\delta_i^\phi$ appearing in the $i$-th position across all permutations is $\frac{1}{T}(\sigma(s) - \sigma(\hat{s}|s))$. We further derive the expected value of the repair score as follows:

$$\mathbb{E}[R(s)] = \sigma(s) - \frac{1}{T+1} \sum_{i=1}^{T} \mathbb{E}\left[\delta_i^\phi\right] \cdot (T - i + 1) = \sigma(s) - \frac{1}{T+1} \cdot \frac{\sigma(s) - \sigma(\hat{s}|s)}{T} \cdot \frac{T(T+1)}{2}. \quad (9)$$

Therefore, as the number of random permutations $N$ approaches infinity, the average repair score converges as follows:

$$\lim_{N \to \infty} \frac{1}{N} \sum_{n=1}^{N} R^{(n)}(s) = \lim_{N \to \infty} \frac{1}{N} \sum_{n=1}^{N} \left( \frac{1}{T+1} \sum_{t=0}^{T} \sigma^{(n)}(s_t|s) \right) = \frac{1}{2}(\sigma(s) + \sigma(\hat{s}|s)). \quad (10)$$

We thus simplify the repair score to $R(s) = \frac{1}{2}(\sigma(s) + \sigma(\hat{s}|s))$, improving detection performance while avoiding intermediate score computations during repair.

## 4 Experiments

### 4.1 Experimental Setup

**Datasets.** To evaluate performance across diverse domains, we collect 4,800 human-written texts from three representative tasks: news article writing (XSum [23]), story generation (WritingPrompts [8]), and academic writing (Arxiv [24]). For each text, we construct task-specific prompts (see Appendix C) and generate corresponding AI outputs using three advanced LLMs: GPT-4 Turbo, Gemini-2.0 Flash, and Claude-3.7 Sonnet. We further sample 2,000 balanced examples from each of three high-quality detection benchmarks—M4 [33], DetectRL [35], and RealDet [41]—to ensure fair and comprehensive evaluation across real-world scenarios.

**Metrics.** We adopt the area under the receiver operating characteristic curve (AUROC [18]) and F1 score to evaluate detection performance, where higher values indicate better separability between human-written and AI-generated texts.

**Baselines.** We compare DNA-DetectLLM with existing training-based and training-free methods. For training-based methods, we include OpenAI-D [28], Biscope [11] and R-Detect [29]. For training-free methods, we consider classic zero-shot detectors including Likelihood [15], LogRank [9], and Entropy [17], along with several recent SOTA approaches such as DetectGPT [22], Fast-DetectGPT [4], Binoculars [14], and Lastde++ [36]. More baseline comparisons are provided in Appendix E.

**Implementation details.** In real-world detection scenarios, the source and distribution of textual data are often unknown, constituting an out-of-distribution (OOD) detection problem. To ensure fairness for training-based methods, we exclusively train on the HC3 dataset [10], which is entirely disjoint from the test sets. For training-free methods, the choice of LLM used for scoring can introduce significant performance variation [3]. To eliminate this factor, we standardize the reference (or scoring) model across all methods by employing Falcon-7B-Instruct [25] to compute token generation probabilities. Moreover, Fast-DetectGPT, Binoculars, Lastde++, and DNA-DetectLLM utilize Falcon-7B [25] as the observer (or sampling) model, while DetectGPT uses T5-3B [26]. During testing, the maximum input token length is capped at 1024. More details are in Appendix D.

### 4.2 Main Results

Table 1 compares the detection performance of DNA-DetectLLM against other baselines across different writing tasks and various generation models. DNA-DetectLLM consistently achieves

Table 1: AUROC (%) of detectors on human-written vs. AI-generated text across datasets and LLMs.

| Detectors | XSum | | | WritingPrompt | | | Arxiv | | | Avg. |
|---|---|---|---|---|---|---|---|---|---|---|
| | GPT-4 Turbo | Gemini-2.0 Flash | Claude-3.7 Sonnet | GPT-4 Turbo | Gemini-2.0 Flash | Claude-3.7 Sonnet | GPT-4 Turbo | Gemini-2.0 Flash | Claude-3.7 Sonnet | |
| **Training-based Methods** | | | | | | | | | | |
| OpenAI-D | 60.51 | 68.93 | 62.96 | 50.94 | 59.47 | 57.28 | 49.63 | 51.40 | 68.57 | 58.85 |
| Biscope | 75.08 | 95.63 | 94.09 | 80.08 | 98.71 | 98.05 | 82.53 | 99.74 | 96.61 | 91.17 |
| R-Detect | 63.56 | 45.63 | 51.13 | 73.58 | 71.07 | 75.74 | 56.47 | 57.24 | 53.55 | 60.89 |
| **Training-free Methods** | | | | | | | | | | |
| Entropy | 72.26 | 54.85 | 74.90 | 87.85 | 90.36 | 91.17 | 45.60 | 79.96 | 80.42 | 75.26 |
| Likelihood | 70.03 | 69.50 | 70.39 | 80.82 | 95.52 | 85.85 | 57.87 | 93.60 | 86.24 | 78.87 |
| LogRank | 69.61 | 69.26 | 69.81 | 78.90 | 94.53 | 84.13 | 58.17 | 94.15 | 85.80 | 78.26 |
| DetectGPT | 61.50 | 68.09 | 61.36 | 72.80 | 89.33 | 78.17 | 56.10 | 92.18 | 90.27 | 74.42 |
| Fast-DetectGPT | 98.33 | 92.54 | 94.30 | 97.79 | 98.58 | 94.14 | 92.12 | 99.80 | 98.21 | 96.20 |
| Binoculars | 98.06 | 95.56 | 96.83 | 97.73 | 99.53 | 97.29 | 93.69 | 99.87 | 97.99 | 97.39 |
| Lastde++ | 97.25 | 90.87 | 92.77 | 95.12 | 98.19 | 92.36 | 90.43 | 99.54 | 97.57 | 94.90 |
| **DNA-DetectLLM** | **99.31** | **96.65** | **98.45** | **98.86** | **99.72** | **98.51** | **95.00** | **99.88** | **98.35** | **98.30** |
| **DNA-DetectLLM with Other Repair Orders** | | | | | | | | | | |
| Low-to-high | 98.94 | 94.98 | 97.31 | 98.29 | 99.41 | 97.77 | 92.48 | 99.75 | 97.91 | 97.43 |
| High-to-low | 99.14 | 97.22 | 98.40 | 98.63 | 99.81 | 98.37 | 94.67 | 99.77 | 97.44 | 98.16 |
| Sequential Repair | 98.90 | 97.82 | 98.37 | 96.80 | 99.77 | 98.55 | 95.78 | 99.91 | 98.21 | 98.23 |

Table 2: Detection performance (AUROC and F1 score) on public benchmark datasets.

| Detectors | M4 | | DetectRL Multi-LLM | | DetectRL Multi-Domain | | RealDet | | Avg. | |
|---|---|---|---|---|---|---|---|---|---|---|
| | AUROC | $F_1$ | AUROC | $F_1$ | AUROC | $F_1$ | AUROC | $F_1$ | AUROC | $F_1$ |
| OpenAI-D | 77.51 | 71.18 | 78.15 | 71.90 | 74.60 | 70.03 | 84.75 | 77.47 | 78.75 | 72.65 |
| Biscope | 79.74 | 73.08 | 79.97 | 73.20 | 76.52 | 71.64 | 92.88 | 86.90 | 82.28 | 76.21 |
| R-Detect | 61.91 | 67.14 | 67.40 | 66.56 | 79.19 | 73.38 | 65.93 | 67.72 | 68.61 | 68.70 |
| Entropy | 83.72 | 79.10 | 64.30 | 71.92 | 47.82 | 69.24 | 75.42 | 74.72 | 67.82 | 73.75 |
| Likelihood | 85.77 | 78.38 | 66.82 | 66.71 | 48.96 | 66.69 | 85.35 | 79.75 | 71.73 | 72.88 |
| LogRank | 87.50 | 80.70 | 67.30 | 66.71 | 50.55 | 66.69 | 86.28 | 80.69 | 72.91 | 73.70 |
| DetectGPT | 73.13 | 70.11 | 49.57 | 66.67 | 34.67 | 66.67 | 78.69 | 73.80 | 59.02 | 69.31 |
| Fast-DetectGPT | 89.77 | 84.12 | 82.26 | 75.93 | 74.98 | 68.91 | 93.25 | 90.00 | 85.07 | 79.74 |
| Binoculars | 90.00 | 87.40 | 83.21 | 82.87 | 77.45 | 80.20 | 93.64 | 90.51 | 86.08 | 85.25 |
| Lastde++ | 91.43 | 84.97 | 75.36 | 69.24 | 67.30 | 66.67 | 93.90 | 89.41 | 82.00 | 77.57 |
| **DNA-DetectLLM** | **91.74** | **87.72** | **88.97** | **84.85** | **88.23** | **84.94** | **94.48** | **90.58** | **90.86** | **87.02** |

state-of-the-art performance under all settings, with an average AUROC of 98.30%, representing a relative improvement of **0.93%**. Specifically, it yields relative gains of **1.36%**, **0.87%**, and **0.58%** on the XSum, WritingPrompts, and Arxiv datasets, respectively, demonstrating strong cross-domain generalization. This strong generalization can be attributed to DNA-DetectLLM's ability to dynamically capture generation discrepancies between domain-specific text and its ideal AI-generated counterpart through the mutation-repair mechanism, enabling robust identification of human-written versus AI-generated text across diverse domains.

Table 2 evaluates the real-world detection performance of all methods on three high-quality public benchmarks. DNA-DetectLLM demonstrates superior reliability, with average AUROC and F1 score improvements of **5.55%** and **2.08%**, respectively. Notably, it achieves significant AUROC gains on the challenging DetectRL settings—**6.92%** on Multi-LLM and **13.92%** on Multi-Domain. This improvement can be attributed to the inherent difficulty of DetectRL, where both positive and negative samples may include mixtures of texts with the same label due to the dataset's construction. Such complexity hinders traditional training-free methods that rely on fixed statistical scores. In contrast, DNA-DetectLLM accurately computes repair scores for intricately constructed input texts by aligning them with their respective ideal AI-generated sequences. This flexible repair-based scoring allows for more accurate detection under distributional overlap and ambiguous cases, underscoring the practical utility of DNA-DetectLLM in complex detection scenarios.

## 4.3 Robustness

### 4.3.1 Robustness against Various Attacks

Figure 4 illustrates the AUROC curves of DNA-DetectLLM and other baselines against various attacks (see Appendix F for details). We conducted experiments on texts generated by GPT-4 Turbo, Gemini-2.0 Flash, and Claude-3.7 Sonnet, incorporating two distinct attacks: token-level edits and paraphrasing. Editing attacks involved random insertion, deletion, or substitution of tokens at rates of

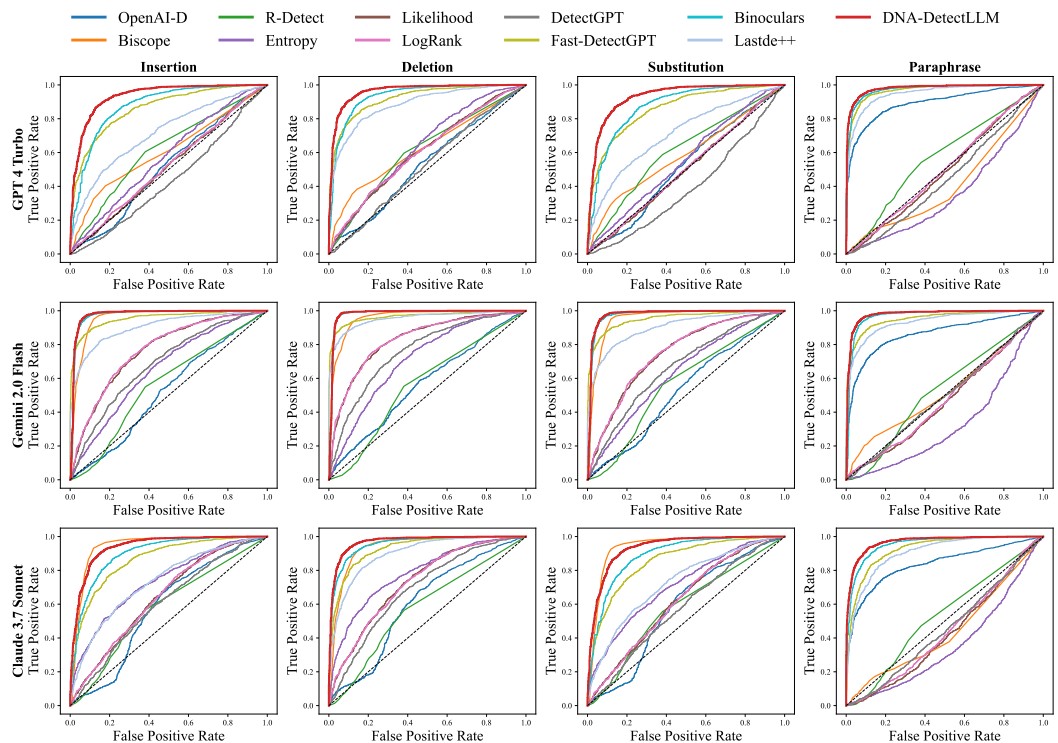

Figure 4: AUROC curves of DNA-DetectLLM and baselines under paraphrasing and editing attacks.

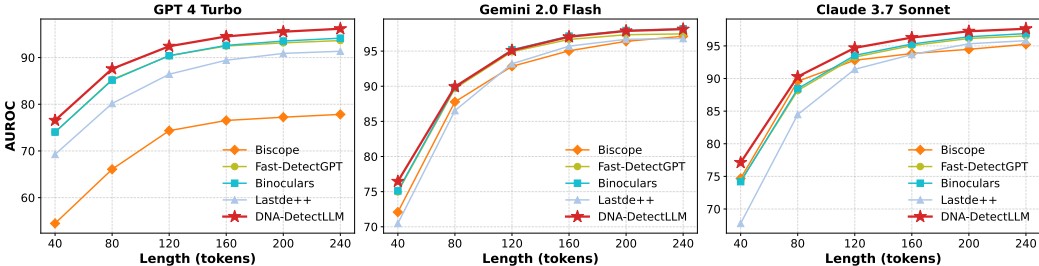

Figure 5: Detection performance (AUROC) on input texts truncated to the target number of tokens.

1%. The paraphrasing attacks employed DIPPER [21] to rephrase AI-generated texts. To maintain clean labels, attacks were exclusively applied to AI-generated texts.

The results demonstrate that DNA-DetectLLM exhibits strong robustness against a variety of adversarial attacks. For instance, on GPT-4 Turbo-generated text, our method achieves relative AUROC improvements of **6.65%**, **3.17%**, **6.62%**, and **0.81%** under insertion, deletion, substitution, and paraphrasing attacks, respectively. Notably, the improvement is particularly pronounced under low false positive rate (FPR) conditions. We attribute this robustness to the observation that although token-level edits are limited in scope, they can substantially alter the generation probability distribution of the input sequence while having minimal impact on its ideal AI-generated sequence. As a result, DNA-DetectLLM is still able to compute accurate repair scores for reliable detection. Moreover, even under paraphrasing attacks using Dipper, the method effectively captures intrinsic deviations from the ideal sequence and maintains a high AUROC of 97.23%. These findings highlight DNA-DetectLLM's capacity to detect adversarially manipulated AI-generated text, even when such attacks are designed to evade detection.

### 4.3.2 Robustness on Different Lengths

Prior research [4, 31] indicates that token length significantly affects detection performance, with shorter texts proving more challenging to detect. We investigate the impact by truncating the input

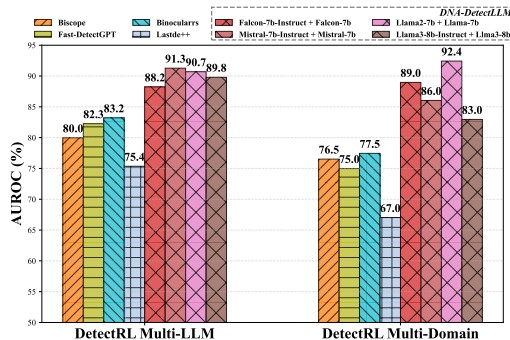

Figure 6: Comparison of DNA-DetectLLM's performance when using different LLM pairs.

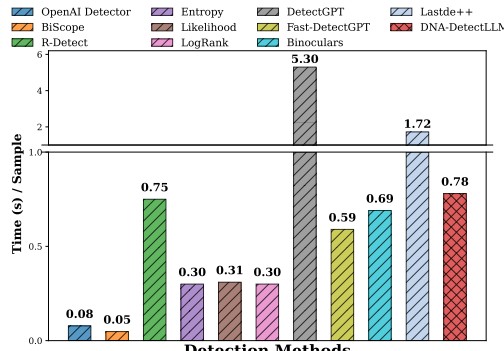

Figure 7: Comparison of time costs for processing a single sample for each method.

texts to various target tokens. Figure 5 presents the detection performance across varying lengths for five methods: DNA-DetectLLM, Binoculars, Fast-DetectGPT, Lastde++, and Biscope. Results show that DNA-DetectLLM consistently outperforms all baselines across varying lengths. On GPT-4 Turbo-generated text, it achieves an average AUROC improvement of **2.46%**. While all methods benefit from longer inputs, DNA-DetectLLM exhibits a greater advantage on shorter texts. At a token length of 40, it surpasses the second-best method by **3.38%**, **1.80%**, and **3.27%**, respectively. These findings suggest that our method enables detection at shorter lengths by extracting more discriminative features from limited textual input.

### 4.4 Ablation Studies

We further evaluated the importance of repair order and different base LLMs through two types of ablation experiments. More detailed ablation studies are available in Appendix G.

**Repair Score-based Detection under Various Repair Orders.** Table 1 compares the detection performance of DNA-DetectLLM under various repair orders. When the repair order is changed to High-to-low, Low-to-high, or Sequential Repair, a slight performance drop is observed. Although these strategies still outperform other baselines, they require recalculating the conditional score after each mutated token repair, resulting in significantly increased computational cost. In contrast, the simplified repair score $(R(s) = \frac{1}{2}(\sigma(s) + \sigma(\hat{s}|s)))$ maintains strong performance while improving efficiency by an order of magnitude, highlighting its necessity in practical deployment.

**DNA-DetectLLM's Performance with Different $M_1$ and $M_2$.** Figure 6 evaluates four different LLM combinations: "Falcon-7B-Instruct + Falcon-7B", "Llama-3-8B-Instruct + Llama-3-8B", "Mistral-7B-Instruct + Mistral-7B", and "Llama-2-7B + Llama-7B". Results demonstrate that any of these combinations significantly outperform existing baselines, with an average performance improvement of 15.28%. Interestingly, the combination "Llama-2-7B + Llama-7B" slightly exceeds the default combination "Falcon-7B-Instruct + Falcon-7B" used in our main experiments, achieving AUROC of 92.4% and 90.7%. These findings highlight the inherent effectiveness of DNA-DetectLLM, suggesting its robust detection performance is not reliant on any specific LLM combination, with potential for further enhancement through better LLM pairings.

### 4.5 Efficiency Analysis

Efficiency is critical for AI-generated text detection, as slow detection speeds hinder large-scale or real-time monitoring in practical scenarios. Figure 7 illustrates the average processing time per sample for each method. To eliminate the confounding factor of text length, we randomly sampled 1,000 long texts from the RealDet dataset, truncated them to 300 tokens, and measured average detection cost with a batch size of 1. We observe that training-based methods such as Biscope and OpenAI-D were the fastest, requiring less than 0.1s per text, but these methods entail significant training overhead. Among training-free methods, classical methods like Likelihood, Logrank, and Entropy are faster, with inference times around 0.3s, but their detection accuracy did not meet our

requirements. DNA-DetectLLM, Binoculars, and Fast-DetectGPT processed each sample in 0.8s, with DNA-DetectLLM achieving the better detection performance.

## 5 Conclusion

In this paper, we introduce DNA-DetectLLM, a novel zero-shot AI-generated text detection method via a DNA-inspired mutation-repair paradigm. Extensive experiments demonstrate that DNA-DetectLLM consistently achieves SOTA detection performance while exhibiting strong robustness across diverse scenarios. We hope our work offers new insights and perspectives for AI-generated text detection and plan to further explore the mutation-repair paradigm to enhance detection performance.

## Acknowledgments

This work is supported by the Postdoctoral Fellowship Program of CPSF under Grant Number GZC20251076, and the National Natural Science Foundation of China (No.U2336202).

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

# A    Limitations

Due to memory constraints, we were unable to scale up the batch size for evaluating method efficiency under ample computational resources. As a result, this study does not fully explore performance differences in real-world scenarios involving large-scale, real-time monitoring of AI-generated texts. The reported efficiency results are based on relative comparisons under a uniform small-batch setting.

# B    Broader Impacts

The proposed DNA-DetectLLM contributes to the field of AI-generated text detection by improving accuracy and introducing a novel detection perspective to the research community. Its enhanced reliability may serve as a reference in socially relevant scenarios where auxiliary judgment is required. However, we emphasize that detection inherently carries a risk of implication or accusation. While our method demonstrates strong performance, we strongly oppose the use of its outputs as direct evidence in punitive or disciplinary contexts. Regardless of its accuracy, such applications could lead to serious consequences and misuse.

# C    Prompt Design for Main Data

Table 3: Examples of input prompts and corresponding outputs across different writing tasks, where the outputs are sampled from GPT-4-generated texts.

| Writing Task | Input Prompt | Output |
|:---:|:---:|:---:|
| News Article Writing | A police source told the BBC that an infiltrator from the Taliban had allowed militants into the police station in the regional capital of Lashkar Gah last night. Please continue. | The militants, believed to be associated with the Taliban, reportedly raided the police station and engaged in a prolonged gun battle with the officers... |
| Story Generation | Two kids entered the Rockmount Zoo. Please continue. | As they excitedly crossed the threshold, their eyes widened at the sight of the vibrantly colored parrots squawking from the treetops... |
| Academic Writing | Please write an abstract based on the following title: "Pure Exploration and Regret Minimization in Matching Bandits". | This paper delves into the field of pure exploration and regret minimization in the context of matching bandits problems - an important area in machine learning... |

Table 3 presents the general-purpose prompts we designed for different writing tasks. Using these prompts, we generated 4,800 AI-generated texts—corresponding to human-written texts—across GPT-4 Turbo, Gemini-2.0 Flash, and Claude-3.7 Sonnet, which were used in our experiments. To ensure reproducibility, we explicitly report the generation parameters for each API call (note that Top-$k$ is not manually configurable):

- **GPT-4 Turbo**: `gpt-4-turbo-2024-04-09`, Temperature = 1.0, Top-$p$ = 1.0.
- **Gemini 2.0 Flash**: `gemini-2.0-flash-001`, Temperature = 1.0, Top-$p$ = 0.95.
- **Claude 3.7 Sonnet**: `claude-3-7-sonnet@20250219`, Temperature = 1.0, Top-$p$ = 1.0.

# D    Main Experiments Supplement

All experiments are conducted on a single NVIDIA A100 GPU with 80GB of memory. Unless otherwise specified, default settings are used for temperature, top-k, and other generation parameters. No additional hyperparameter tuning is involved in this study. For training-based methods (Biscope and R-Detect), models are trained on 4,000 balanced samples from the HC3 dataset, and the best-performing checkpoints are selected based on validation performance on a separate 2,000-sample

validation set. In the main experiments, all reported F1 scores are the maximum values obtained by selecting the optimal threshold based on the positive and negative scores of each method.

Table 4: Comparison of F1 score across public benchmark datasets.

| Method | M4 | DetectRL Multi-LLM | DetectRL Multi-Domain | RealDet | Avg. |
|---|---|---|---|---|---|
| OpenAI-D | 68.01 | 70.55 | 67.89 | 70.41 | 69.22 |
| Biscope | 71.75 | 72.00 | 68.91 | 81.23 | 73.97 |
| R-Detect | 67.14 | 66.56 | 66.26 | 67.55 | 66.88 |
| Entropy | 75.39 | 70.10 | 55.04 | 60.33 | 65.22 |
| Likelihood | 66.82 | 66.62 | 66.60 | 66.87 | 66.73 |
| LogRank | 66.80 | 66.69 | 66.60 | 66.82 | 66.73 |
| DetectGPT | 54.53 | 42.41 | 30.79 | 66.59 | 48.58 |
| Fast-DetectGPT | 81.27 | 75.84 | 68.06 | 84.72 | 77.47 |
| Binoculars | 84.82 | 80.97 | 76.24 | 82.15 | 81.05 |
| Lastde++ | 82.74 | 69.17 | 61.72 | 84.59 | 74.56 |
| **DNA-DetectLLM** | **85.15** | **84.49** | **83.94** | **84.72** | **84.58** |

To ensure a fairer performance comparison, we select fixed thresholds for all methods based on scores computed on a separate clean dataset (e.g., DNA-DetectLLM: 0.6533, Binoculars: 0.9366, etc.). Subsequently, we recompute and report the F1 scores in Table 4 using these fixed thresholds. The clean dataset used for threshold selection consists of over 3,000 samples generated by GPT-4, Gemini, and Claude based on human-written texts sourced from XSum, WritingPrompt, and Arxiv.

# E   Additional Performance Comparisons

Table 5: Comparison of AUROC (%) across benchmark datasets.

| Method | XSum | WritingPrompt | Arxiv | PubMedQA | Avg. |
|---|---|---|---|---|---|
| Revise-Detect | 39.73 | 65.54 | 95.31 | – | 66.86 |
| GECScore | 70.84 | 66.31 | 64.91 | – | 67.35 |
| DNA-GPT | 65.46 | 75.22 | 70.13 | 82.32 | 73.28 |
| ImBD | 88.07 | 93.06 | 91.06 | 92.59 | 91.20 |
| GPTZero | 99.01 | 98.54 | 94.42 | 88.48 | 95.11 |
| **DNA-DetectLLM** | **99.31** | **98.86** | **95.00** | **97.08** | **97.56** |

As shown in Table 5, we expanded our comparative experiments to include additional baselines—Revise-Detect [40], GECScore [34], DNA-GPT [38], IMBD [6] (recent but non–state-of-the-art methods), and GPTZero (a widely used commercial detector). We also incorporated results on the biomedical short-text dataset PubMedQA [19], which further demonstrate the strong and consistent performance of DNA-DetectLLM across diverse domains and detection settings.

# F   Robustness Experiments

In this study, we do not consider adversarial attacks on human-written texts, as evasion in such cases is generally inconsequential. Instead, we focus on adversarial scenarios involving AI-generated texts, introducing two common attack types: paraphrasing and token-level editing. For paraphrasing attacks, we employ DIPPER with hyperparameters set to a lexical diversity of 60 and a syntactic diversity of 60. This level of paraphrasing is sufficient to potentially bypass SOTA detectors. For editing attacks, we tokenize the input using the GPT-2 tokenizer and apply random insertions, deletions, and substitutions to 1% of the tokens. The inserted or substituted tokens are sampled uniformly from the tokenizer's vocabulary.

Table 6, Table 7, and Table 8 report the detection performance of all methods under various adversarial attacks across different AI-generated texts. Notably, DNA-DetectLLM consistently achieves strong performance across all adversarial scenarios, demonstrating robustness to both editing and paraphrasing attacks. In contrast, training-free methods are significantly affected by token-level edits, while training-based methods are more vulnerable to paraphrasing-based attacks.

Table 6: AUROC (%) for GPT-4 Turbo against various attacks.

| Method | Insertion | Deletion | Substitution | Paraphrase |
|---|---|---|---|---|
| OpenAI-D | 51.54 | 52.95 | 51.62 | 89.50 |
| Biscope | 60.75 | 62.66 | 58.59 | 41.16 |
| R-Detect | 61.71 | 61.27 | 61.08 | 57.49 |
| Entropy | 57.52 | 63.38 | 55.44 | 36.04 |
| Likelihood | 51.54 | 60.69 | 49.89 | 49.33 |
| LogRank | 52.19 | 60.69 | 50.45 | 50.48 |
| DetectGPT | 43.78 | 52.36 | 39.67 | 44.60 |
| Fast-DetectGPT | 86.16 | 92.05 | 85.89 | 96.70 |
| Binoculars | 87.55 | 93.32 | 87.28 | 97.23 |
| Lastde++ | 71.60 | 88.51 | 71.34 | 94.94 |
| DNA-DetectLLM | **93.37** | **96.28** | **93.06** | **98.02** |

Table 7: AUROC (%) for Gemini-2.0 Flash against various attacks.

| Method | Insertion | Deletion | Substitution | Paraphrase |
|---|---|---|---|---|
| OpenAI-D | 52.28 | 56.72 | 53.02 | 86.46 |
| Biscope | 95.48 | 95.79 | 95.15 | 52.61 |
| R-Detect | 56.40 | 56.37 | 56.84 | 52.97 |
| Entropy | 65.59 | 71.11 | 63.47 | 33.50 |
| Likelihood | 76.41 | 82.86 | 74.73 | 47.02 |
| LogRank | 76.79 | 82.93 | 74.98 | 47.65 |
| DetectGPT | 68.74 | 76.01 | 66.60 | 49.68 |
| Fast-DetectGPT | 95.39 | 96.57 | 95.23 | 95.27 |
| Binoculars | 97.41 | 98.06 | 97.35 | 96.78 |
| Lastde++ | 90.39 | 95.31 | 90.16 | 93.41 |
| DNA-DetectLLM | **97.72** | **98.16** | **97.69** | **97.80** |

Table 8: AUROC (%) for Claude-3.7 Sonnet against various attacks.

| Method | Insertion | Deletion | Substitution | Paraphrase |
|---|---|---|---|---|
| OpenAI-D | 57.99 | 60.50 | 57.80 | 83.62 |
| Biscope | **93.99** | 93.47 | **93.65** | 44.34 |
| R-Detect | 58.44 | 58.24 | 58.06 | 53.02 |
| Entropy | 73.32 | 78.29 | 70.87 | 35.93 |
| Likelihood | 64.11 | 73.01 | 61.93 | 42.80 |
| LogRank | 64.26 | 72.79 | 61.99 | 44.21 |
| DetectGPT | 60.39 | 68.81 | 57.54 | 45.37 |
| Fast-DetectGPT | 86.38 | 92.74 | 85.34 | 92.56 |
| Binoculars | 89.81 | 95.34 | 88.88 | 95.33 |
| Lastde++ | 73.47 | 89.85 | 72.54 | 90.43 |
| DNA-DetectLLM | 93.77 | **96.89** | 93.25 | **96.81** |

Table 9: F1 score (%) under paraphrasing and editing attacks

| Method | GPT-4 Turbo | | | | Gemini-2.0 Flash | | | | Claude-3.7 Sonnet | | | | Avg. |
|---|---|---|---|---|---|---|---|---|---|---|---|---|---|
| | Insert | Deletion | Substitution | Paraphrase | Insert | Deletion | Substitution | Paraphrase | Insert | Deletion | Substitution | Paraphrase | |
| OpenAI-D | 57.17 | 58.23 | 58.91 | 74.24 | 59.52 | 60.24 | 60.34 | 72.99 | 65.00 | 66.04 | 65.15 | 71.93 | 64.15 |
| Biscope | 54.31 | 55.19 | 50.76 | 27.09 | 86.46 | 86.34 | 86.30 | 42.42 | 86.09 | 85.80 | 85.80 | 32.73 | 64.94 |
| R-Detect | 66.46 | 66.46 | 66.46 | 66.50 | 66.16 | 66.16 | 66.16 | 66.16 | 66.63 | 66.63 | 66.63 | 66.63 | 66.42 |
| Entropy | 62.53 | 65.04 | 61.92 | 56.82 | 65.95 | 68.69 | 64.68 | 57.09 | 68.27 | 71.36 | 67.02 | 57.20 | 63.88 |
| Likelihood | 66.65 | 66.65 | 66.65 | 66.61 | 66.54 | 66.54 | 66.54 | 66.54 | 66.69 | 66.69 | 66.69 | 66.65 | 66.62 |
| LogRank | 66.61 | 66.61 | 66.61 | 66.61 | 66.57 | 66.57 | 66.57 | 66.50 | 66.65 | 66.65 | 66.65 | 66.65 | 66.60 |
| DetectGPT | 21.80 | 30.87 | 17.13 | 21.04 | 54.86 | 65.84 | 52.22 | 30.18 | 43.03 | 54.86 | 38.79 | 22.13 | 37.73 |
| Fast-DetectGPT | 77.10 | 84.28 | 76.24 | 89.38 | 87.90 | 89.04 | 87.85 | 88.03 | 77.00 | 85.22 | 75.81 | 84.80 | 83.55 |
| Binoculars | 81.33 | 87.16 | 81.15 | 91.50 | 92.44 | 93.08 | 92.28 | 91.65 | 83.15 | 88.71 | 82.36 | 88.78 | 87.80 |
| Lastde++ | 60.86 | 80.67 | 60.39 | 87.33 | 82.46 | 87.16 | 81.88 | 85.87 | 61.46 | 82.22 | 61.26 | 83.02 | 76.22 |
| **DNA-DetectLLM** | **86.74** | **90.36** | **86.01** | **93.09** | **94.22** | **94.94** | **93.91** | **93.63** | **87.28** | **91.58** | **86.81** | **91.06** | **90.80** |

Table 9 reports the F1-score performances for the primary robustness experiments. Notably, DNA-DetectLLM consistently exhibits superior robustness compared to other baselines in practical scenarios.

# G   Ablation Study Supplement

Table 10: Ablation results across different repair strategies and datasets.

| Setting | XSum | WP | Arxiv | M4 | RealDet | Avg. | Time Cost (s) |
|---------|------|------|-------|------|---------|------|---------------|
| Default | 98.14 | 99.03 | 97.74 | 91.74 | 94.48 | **96.23** | **0.78** |
| Low-to-High | 97.08 | 98.49 | 96.71 | 92.42 | 94.71 | 95.88 | 14.11 |
| High-to-Low | 98.25 | 98.94 | 97.29 | 89.56 | 93.59 | 95.53 | 14.45 |
| Sequential Repair | 98.36 | 98.37 | 97.97 | 91.26 | 93.71 | 95.93 | 14.55 |

Table 10 compares the detection performance and inference time of DNA-DetectLLM under different repair orders. The results show that the default setting achieves superior performance compared to alternative strategies, while reducing computation time by nearly 20×, highlighting its practical efficiency.

