# OpenReview forum: "DNA-DetectLLM: Unveiling AI-Generated Text via a DNA-Inspired Mutation-Repair Paradigm"
_NeurIPS.cc/2025/Conference — NeurIPS 2025 spotlight_

### Official Review · Reviewer_YKTY · 2025-06-28

**Clarity:** 3
**Significance:** 3
**Originality:** 3
**Rating:** 4
**Confidence:** 4

**Summary:**

This paper is inspired by the DNA repair process in biology. The paper proposes a new way to detect AI-generated text: iteratively “repairing” a text sequence and measuring how hard it is to make it match the ideal AI-generated version. Then, the paper proposes the difficulty of the repair process of fixing fraction quantification. We introduced one to capture the cumulative effort required to complete the conversion. Last, by comparing the repair score with the calibrated threshold, DNA-DetectLLM robustly distinguishes AI-generated text and human-written text, using the basic differences that deviate from the ideal generation.

**Questions:**

Please See Weaknesses.

**Ethical Concerns:**

["NO or VERY MINOR ethics concerns only"]

**Final Justification:**

I have resolved my doubts. I gave this paper a score of 4 because it performs very well on the current benchmarks, and compared to the previous method, DNA-GPT, it has achieved a significant improvement in speed, which is important.

**Limitations:**

Yes

**Paper Formatting Concerns:**

The paper has no formatting issues.

**Quality:**

3

**Strengths And Weaknesses:**

**Strengths**

- The paper is well written and easy to follow.
- The idea is novel and motivated by a strong analogy from biology.
- The experiments are well-designed and show strong results.
- The baselines used for comparison are mostly appropriate and fairly broad.

**Weaknesses**

1. The datasets used in the main experiments (XSum, Writing, ArXiv) are quite similar—they are all long-form writing tasks. A more challenging dataset, such as **PubMedQA**, which is used in Fast-DetectGPT and focuses on scientific question answering, would be helpful to test how well the method works in a different domain.

2. It would be useful to include **GPTZero**, a popular commercial detection tool, as a baseline. This would help readers understand how the method compares to widely-used systems.

3. The authors should include comparisons with more recent and stronger baselines:
- **DNA-GPT** (ICLR 2024)
- **Text Fluoroscopy** (EMNLP 2024)
- **ImBD** (AAAI 2025)
- **Biscope** (NeurIPS 2024)

4. The paper mentions using outputs from GPT-4 Turbo, Gemini 2.0 Flash, and Claude 3.7 Sonnet. However, it does not provide details such as the **temperature** or **model versions** used to generate these outputs. This information is important for reproducibility. Also, since GPT-4 Turbo is now being phased out, the authors could consider using GPT-4o for future experiments.

---

> ### Author Rebuttal · Authors · 2025-07-26
>
> Dear Reviewer,
>
> We sincerely appreciate the time and effort you have dedicated to reviewing our manuscript and providing valuable suggestions. In response to your feedback, we offer the following point-by-point replies:
>
> **R1 (to W1 & W3): On Dataset and Baseline Expansions**
>
> To further evaluate generalization performance, we have included results on **PubMedQA**, a more challenging biomedical short-text dataset, in Table 1.
>
> Our experiments already incorporate several of the most recent and state-of-the-art baselines, such as **Biscope** (NeurIPS 2024), **R-Detect** (ICLR 2025), and **Lastde++** (ICLR 2025). In response to your suggestion, we have additionally included several relevant recent methods (e.g., DNA-GPT and ImBD) in Table 1 for a more comprehensive comparison. Note that **DNA-GPT** uses the same scoring model (Falcon-7B-Instruct) as the other methods, and **ImBD** is pretrained on the HC3 dataset.
> As shown in Table 1, **DNA-DetectLLM** maintains strong detection performance on PubMedQA (AUROC = **97.08%**), clearly demonstrating its superior generalization ability across diverse distributions.
>
> **Table 1. AUROC(%) on GPT-4-generated Texts.**
> | Method           | XSum  | WritingPrompt | Arxiv | PubMedQA |
> |------------------|--------|----------------|--------|-----------|
> | ***Biscope***          | -      | -              | -      | 92.39     |
> | ***Fast-DetectGPT***   | -      | -              | -      | 96.31     |
> | ***Binoculars***       | -      | -              | -      | 96.48     |
> | ***Lastde++***         | -      | -              | -      | 87.41     |
> | ***DNA-GPT***          | 65.46  | 75.22          | 70.13  | 82.32     |
> | ***ImBD***             | 88.07  | 93.06          | 91.06  | 92.59     |
> | ***DNA-DetectLLM***    | **99.31**  | **98.86**          | **95.00**  | **97.08**     |
>
> **R2 (to W2): On the Use of GPTZero as a Baseline**
>
> GPTZero is a popular commercial detection tool that was included as a baseline in earlier works such as Fast-DetectGPT. However, many subsequent studies have shown that GPTZero performs significantly less reliably than strong zero-shot methods. Recent state-of-the-art approaches (e.g., **Biscope**, **Lastde++**) have also opted not to include GPTZero in comparisons, instead focusing on benchmarking against more advanced methods. Additionally, the inference cost of GPTZero is relatively high.
> Given these considerations, we believe that including GPTZero as a baseline in our work is not essential.
>
> **R3 (to W4): On the Use of GPT-4 Turbo, Gemini 2.0 Flash, and Claude 3.7 Sonnet**
>
> The model IDs for the text generators used are as follows:
>
> * **GPT-4 Turbo**: `gpt-4-turbo-2024-04-09`
>
> * **Gemini 2.0 Flash**: `gemini-2.0-flash-001`
>
> * **Claude 3.7 Sonnet**: `claude-3-7-sonnet@20250219`
>
> All generations were performed using the default official parameter settings provided by the respective APIs. We will include further implementation details and code snippets in the Appendix of the final version to enhance the reproducibility of our results.
> We once again thank you for your constructive suggestions and hope that our responses have adequately addressed your concerns.
>
> Best regards,
>
> All authors

---

> > ### Comment · Reviewer_YKTY · 2025-08-03
> >
> > **R2 (to W2).** In your response, you argue that ' Given these considerations, we believe that including GPTZero as a baseline in our work is not essential.' **However, this reasoning seems problematic**:
> >
> > (1) The comparisons you cite regarding GPTZero’s inferior performance are based on older versions. GPTZero has undergone multiple updates, and its current performance may differ considerably. Simply stating that “GPTZero performs significantly worse than many strong zero-shot detectors” seems overly assertive and insufficiently supported.
> >
> > (2) You mentioned that 'Recent state-of-the-art approaches (e.g., Biscope, Lastde++) have also opted not to include GPTZero in comparisons', **However, this is inaccurate.** For example, in Biscope’s original paper, GPTZero was explicitly included as a baseline (see Figure 5), and Section 2 (“Background and Related Work”) mentions: “as well as with the most renowned commercial detection API, GPTZero. We surpass these baseline methods in both effectiveness and efficiency.” This contradicts your claim.
> >
> > **If the primary reason for excluding GPTZero is cost, that is understandable. But dismissing it solely on the grounds of “poor performance” without updated evidence, especially when prior works have included it, does not seem well justified. Moreover, such statements could mislead readers if not corrected.**
> >
> > ---
> >
> > **R3 (to W4).**
> > In addition, you stated that “All generations were performed using the default official parameter settings provided by the respective APIs.”
> >
> > However, this statement is insufficiently detailed. For proper reproducibility and for assessing the fairness of your comparisons, you should explicitly list the exact parameter values used for each API (e.g., temperature, top_p, etc.) rather than referring to them vaguely as “default.”
> >
> > In particular, previous works typically set the temperature to 0.7–0.8, whereas OpenAI’s current default temperature is 1.0. Using 1.0 could affect generation style and detection difficulty. Could you clarify whether using 1.0 is intentional and justified?
> >
> > Moreover, since you used multiple different APIs (OpenAI, Gemini, etc.), were the parameters consistent across different models?

---

> > > ### Author Response · Authors · 2025-08-04
> > >
> > > Dear Reviewer：
> > >
> > > We sincerely appreciate your patient response and valuable clarification.In response to your concerns, we provide the following detailed reply:
> > >
> > > **R2 (to W2). Including GPTZero for a More Comprehensive Evaluation of DNA-DetectLLM**
> > >
> > > We apologize for our earlier misunderstanding regarding GPTZero, and we sincerely appreciate your clarification on this matter. Your feedback has helped us recognize that GPTZero is a continually evolving commercial detection tool, whose performance may significantly differ from earlier reported results in the literature. Indeed, recent studies have included GPTZero as a baseline method (we missed it in Biscope), underscoring the importance of comparing it to DNA-DetectLLM. To address this, we have now included a comparison of GPTZero's performance against DNA-DetectLLM on GPT-4-generated texts (including PubMedQA). Table 1 below shows the results. The GPTZero used here is the latest version, with detailed information as follows:
> > > * **GPTZero**: `'version': '2025-07-23-base', 'neatVersion': '3.6b', 'multilingual': False`
> > >
> > > We found that GPTZero exhibits strong competitive performance on longer text datasets (XSum, WritingPrompt, Arxiv), clearly surpassing typical zero-shot detection methods. Conversely, in shorter text scenarios such as PubMedQA, GPTZero achieves a relatively lower AUROC of 88.48%, underperforming certain zero-shot detection methods. In comparison, DNA-DetectLLM consistently demonstrates superior detection performance across diverse data, highlighting its improved generalizability over GPTZero.
> > >
> > > **Table 1. AUROC(%) on GPT-4-generated Texts.**
> > > | Method           | XSum  | WritingPrompt | Arxiv | PubMedQA | Avg.|
> > > |------------------|--------|----------------|--------|-----------|-|
> > > | ***GPTZero***    | 99.01  | 98.54          | 94.42  | 88.48     | 95.11
> > > | ***DNA-DetectLLM***    | **99.31**  | **98.86**          | **95.00**  | **97.08**     | **97.56**
> > >
> > > **R3 (to W4). Detailed API Parameter Declaration for Proper Reproducibility**
> > >
> > > To ensure accurate reproducibility, we explicitly state the generation parameters used for each API call (Top-k is not manually configurable):
> > > * **GPT-4 Turbo**: `gpt-4-turbo-2024-04-09, Temperature=1.0, Top-P=1.0`
> > > * **Gemini 2.0 Flash**: `gemini-2.0-flash-001, Temperature=1.0, Top-P=0.95`
> > > * **Claude 3.7 Sonnet**: `claude-3-7-sonnet@20250219, Temperature=1.0, Top-P=1.0`
> > >
> > > As noted in POGER [1], **an excessively high temperature can significantly degrade the performance of detection methods.** However, in real-world scenarios, users often seek more diverse and natural outputs and therefore do not reduce the default temperature of 1.0. Furthermore, potential malicious users may deliberately exploit high-temperature outputs to evade detection systems. For these reasons, we believe that using 'Temperature = 1.0' better reflects the true difficulty of AI-generated text detection in practice.
> > >
> > > To further enhance reproducibility, we will include the prompt designs and API call examples in the appendix of the final manuscript.
> > >
> > > We sincerely thank you for your thoughtful and constructive suggestions. We truly hope that our responses have meaningfully addressed your concerns, and we would greatly appreciate the opportunity for further interactions with you.
> > >
> > > [1] Ten Words Only Still Help: Improving Black-Box AI-Generated Text Detection via Proxy-Guided Effcient Re-Sampling. IJCAI, 2024.
> > >
> > > Best regards,
> > >
> > > All authors

---

> ### Comment · Reviewer_YKTY · 2025-08-04
>
> I appreciate the authors’ clarification and am glad to see that the misunderstanding regarding GPTZero has now been resolved.
> While I originally mentioned that considering cost issues, it would be acceptable not to compare with GPTZero, what is unacceptable are the authors’ unfounded or incorrect assertions, such as claiming that BiScope does not include GPTZero, or arguing that 'However, many subsequent studies have shown that GPTZero performs significantly less reliably than strong zero-shot methods.'
>
> **Furthermore, there are still some assertions in the authors’ reply that leave me very confused, for example: 'However, in real-world scenarios, users often seek more diverse and natural outputs and therefore do not reduce the default temperature of 1.0.'**
>
> **I do not think this statement is accurate.** First, for GPT models, a temperature of 1.0 is actually quite high. While a higher temperature may lead to more creative text, it can also increase hallucinations. In real-world scenarios, users who pursue high-quality text (e.g., with fewer hallucinations) may reduce the temperature. Considering the domains used in text detection—such as XSum and writing—higher creativity may be desirable. Thus, balancing text quality, naturalness, and creativity, a temperature around 0.7–0.8 is often more reasonable. In fact, this is also the temperature used in Fast-DetectGPT and DNA-GPT.
>
> That said, some papers do use a temperature of 1, such as DetectGPT and DetectRL, which mention using it to “promote the generation of creative and unpredictable text.” So, if the intention is to increase the difficulty of detection, this is indeed reasonable.
>
> However, making a direct assertion like 'in real-world scenarios, users often seek more diverse and natural outputs and therefore do not reduce the default temperature of 1.0' could be misleading. In fact, in this context, “natural” and a temperature of 1.0 seem somewhat contradictory, as lower temperatures often lead to more natural text.
>
> **The authors do not need to regenerate texts using other temperature settings for experiments. My intention here is simply to correct this statement, because such unfounded assertions can mislead readers. Given the broad real-world applications of text detection, your readers may include practitioners who focus on applications rather than research. In such cases, statements like this could easily lead to misunderstandings.**
>
> Lastly, I appreciate the authors’ paper mainly for its promising experimental results and relatively short time consumption. In fact, the idea of using DNA-inspired methods for detectors can indeed be traced back to DNA-GPT. However, DNA-GPT suffers from extremely long processing times and unsatisfactory results (as reported in many papers). In contrast, your results show both low time cost and effective performance, which is impressive and clearly beneficial for future research. By this, I mean that your approach can serve as a competitive baseline.

---

> > ### Author Response · Authors · 2025-08-04
> >
> > Dear Reviewer:
> >
> > Our statement regarding the temperature setting may have reflected certain personal biases, and we deeply appreciate the opportunity to engage in this meaningful discussion with you.
> >
> > **Finally, we sincerely thank you for the time and effort you have devoted to improving this manuscript. We are also truly grateful to see your recognition of our work.**
> >
> > Best regards,
> >
> > All authors

---

> > > ### Comment · Reviewer_YKTY · 2025-08-09
> > >
> > > I have now resolved my doubts and will keep my score, which is already positive. At the same time, I have also raised
> > > Significance: 2 (fair)
> > > Originality: 2 (fair)
> > > to 3.

---

### Official Review · Reviewer_LFW4 · 2025-06-30

**Clarity:** 3
**Significance:** 3
**Originality:** 4
**Rating:** 5
**Confidence:** 4

**Summary:**

The article proposes a novel method called DNA-DetectLLM, which adopts a perspective inspired by DNA repair to directly and interpretably analyze the intrinsic differences between human-written and AI-generated texts. This method constructs an ideal AI-generated sequence for each input, iteratively repairs suboptimal tokens, and quantifies the cumulative repair effort as an interpretable detection signal.

**Questions:**

I am curious about the experimental setup for the ablation studies. Is it in-distribution or out-of-distribution? For example, in terms of text length, was the model trained on texts of the same length and then tested on texts of the same length?

**Ethical Concerns:**

["NO or VERY MINOR ethics concerns only"]

**Final Justification:**

Given that my major concerns have been addressed and the paper now presents clearly justified, robust, and interesting results, I believe it is suitable for acceptance. I encourage the authors to further clarify the limitations of the supervised detector comparisons in the final version to avoid any misinterpretation. On balance, the paper makes a meaningful contribution to the research community, and I recommend acceptance.

**Quality:**

3

**Strengths And Weaknesses:**

**Strengths**
- The proposed method is both interesting and well-motivated.
- The evaluation of the method is robust, considering mainstream and challenging detection scenarios, with necessary ablation studies conducted.

**Weaknesses**
- The authors seem to have a misunderstanding of the current state of research on detection methods. For example, in line 30, it is stated that “training-free approaches leverage intrinsic statistical differences to distinguish human-written and AI-generated texts,” but in reality, these methods are not entirely “training-free.” They also require extracting feature scores on a training set, which are then used for detection during the testing phase. It would be beneficial to revise the corresponding statements with reference to [1].
- Following the previous weakness, it appears that the authors did not separate the training set from the testing set, leading to unfair comparisons in Table 1 and the related ablation studies. For a fair comparison, training-based methods and training-free methods should be trained on the same training set and evaluated on the same testing set.

Overall, this is a good paper, but there seems to be a significant issue with the experimental setup. If this issue can be resolved, I would be happy to recommend accepting this paper.

**Other Suggestion**

In the related work section, the authors discuss the differences between their approach and previous studies. However, I believe there are still some similar works that have not been mentioned. Overall, the paper assumes that human-written text resembles a mutated chain, where token selection deviates from the optimal probability, resulting in measurable differences. It is worth noting that some works also adopt similar ideas, such as Revise-Detect. [2], which allows LLMs to rewrite the text themselves to fix tokens that deviate from the optimal probability, and GECScore [3], which introduces grammatical errors and LLM's correction preferences to measure the sequence differences between the two types of text. The key difference is that these works operate at the sequence level for "repair" and "correction," whereas the proposed method in this paper performs iterative "repair" and "correction" at the token level. I speculate that the token-by-token repair process may provide more controllable and fine-grained measurements of the deviation patterns between the two types of text. Therefore, comparing this work with such methods would be beneficial in highlighting the advantages of the approach presented in the paper.

[1] A Survey on LLM-Generated Text Detection: Necessity, Methods, and Future Directions. Computational Linguistics, 2025.

[2] Beat LLMs at Their Own Game: Zero-Shot LLM-Generated Text Detection via Querying ChatGPT. EMNLP, 2023.

[3] Who Wrote This? The Key to Zero-Shot LLM-Generated Text Detection Is GECScore. COLING, 2025.

---

> ### Author Rebuttal · Authors · 2025-07-26
>
> Dear Reviewer,
>
> We sincerely appreciate the time and effort you have dedicated to reviewing our manuscript and providing valuable suggestions. In response to your feedback, we offer the following point-by-point replies:
>
> **R1 (to Weakness): Training-free Methods Require No Threshold Training under AUROC Evaluation.**
>
> We understand your point regarding training-free methods not being entirely "training-free." For example, methods like GECScore [1], aside from using predefined thresholds, can also optimize a distribution-specific "optimal threshold" based on training data. Similarly, other training-free methods also require identifying an “optimal threshold” based on feature scores from training data to enhance classification performance in practical applications. However, it is equally important to emphasize that when these methods calculate statistical scores based on relative differences, they indeed do not require additional training. For instance, Fast-DetectGPT [2] calculates log conditional probability differences across multiple samples using sampling and scoring models (LLMs in evaluation mode).
>
> In our experiments, we primarily adopt AUROC, a threshold-independent evaluation metric. Therefore, none of the training-free methods in our experiments involves training on data within the same distribution to identify an "optimal threshold." Instead, they calculate statistical scores for positive and negative samples and directly convert these scores into AUROC values, enabling a fair comparison across methods. Meanwhile, the training-based methods are trained on the general HC3 dataset [3] to simulate real-world out-of-distribution (OOD) scenarios (see Section 4.1), providing a relatively fair evaluation against the zero-shot methods that require no additional training. We hope this clarification addresses your concerns regarding our training and evaluation.
>
> P.S. The computation of F1 scores presented in Table 2 is explained in Appendix D of the manuscript.
>
> **R2 (to Suggestion):** Thank you for your suggestion and for highlighting additional related work. We will expand Section 2 to include a discussion of sequence-level revise or rewrite-based approaches, such as Revise-Detect [4] and GECScore, which assess textual differences through revision-based strategies. As shown in the supplementary Table 1 below, we evaluated the detection performance of Revise-Detect and GECScore on GPT-4-generated texts. Our results show that these methods display substantial performance variability across different data distributions and are considerably less reliable than DNA-DetectLLM. Interestingly, Revise-Detect performs well on the Arxiv dataset. We believe this may be due to the formal writing style of the dataset, where the LLM used for revision (GPT-4o) can more effectively capture the homology of GPT-4-generated texts. To more fully demonstrate the advantages of DNA-DetectLLM, we will include these experimental results in the Appendix of the final version.
>
> **Table 1. AUROC(%) on GPT-4-generated Texts.**
> | Method          | Xsum | WritingPrompt | Arxiv | Avg.   |
> |-----------------|------------------|-------------------|-------------------|--------|
> | ***Revise-Detect***   | 39.73            | 65.54             | 95.31             | 66.86  |
> | ***GECScore***       | 70.84            | 66.31             | 64.91             | 67.35  |
> | ***DNA-DetectLLM***   | 99.31            | 98.86             | 95.00             | **97.72**  |
>
> **R3 (to Question):** The training-free methods do not require any additional training and compute AUROC scores directly based on statistical scoring. In contrast, training-based methods such as Biscope are trained on the HC3 dataset and do not incorporate any explicit handling of input text length. In the robustness experiment on text length, we only truncate the input text during the testing phase, without modifying the training process.
>
> [1] Who Wrote This? The Key to Zero-Shot LLM-Generated Text Detection Is GECScore. COLING, 2025.
>
> [2] Fast-DetectGPT: Efficient Zero-Shot Detection of Machine-Generated Text via Conditional Probability Curvature. ICLR, 2024.
>
> [3] How Close is ChatGPT to Human Experts? Comparison Corpus, Evaluation, and Detection. 2023.
>
> [4] Beat LLMs at Their Own Game: Zero-Shot LLM-Generated Text Detection via Querying ChatGPT. EMNLP, 2023.
>
> Best regards,
>
> All authors

---

> ### Comment · Reviewer_LFW4 · 2025-08-02
> **Official Comment by Reviewer LFW4**
>
> Thank you very much for your clarifications regarding the evaluation procedure with AUROC. I realize that my earlier comment concerning “training” may have been ambiguous—I was specifically referring to the process of selecting an optimal decision threshold for F1-score computation, rather than model training in the conventional sense. I apologize for any confusion this may have caused.
>
> To further clarify my concerns:
>
> **1. Threshold Selection in Table 2**
>
> In Table 2, the reported F1-scores for each method require setting a classification threshold. Could you please elaborate on how this threshold was determined? Specifically, was it selected by optimizing on the test set, or by using a separate "train/statistics" dataset?
>
> If the threshold was chosen directly on the test set (i.e., by selecting the value that yields the highest F1-score on the test data), this may lead to overly optimistic (and potentially overfitted) F1-scores, which could affect the fairness of the comparison, especially for methods that are statistical or “training-free.”
>
> For a fair comparison, it would be preferable if all methods used a threshold selected via a "train/statistics" dataset, and then reported the F1-score on the test set using that threshold. If the threshold was indeed selected on the test set, I would recommend revising the evaluation to align with standard best practices; otherwise, the reported F1-scores might not fully reflect true performance in a real deployment scenario.
>
> **2. F1-score Reporting in Robustness/Ablation Experiments**
>
> I also noticed that the ablation (robustness) experiments currently report only AUROC. While AUROC is a useful metric, I am concerned that it may not fully reflect the practical robustness of the method in real-world applications.
>
> In practice, after model training or statistical analysis and validation, a fixed decision threshold (e.g., optimized on a "train/statistics" dataset or determined by statistical criteria) is set, and this same threshold is used for classifying all new samples—including paraphrased/edited or length-altered examples.
>
> It is important to note that a high AUROC under attack or distribution shift does not always guarantee robust performance with a fixed threshold. For example, under adversarial or distributional shifts, the score distributions may change in such a way that the fixed threshold results in much lower precision, recall, or F1-score, even if the AUROC remains high.
>
> Therefore, I would kindly suggest reporting the F1-score (or other threshold-dependent metrics) in the ablation/robustness experiments as well, using a threshold chosen on a clean (in-distribution) "train/statistics" dataset, and applying it directly to the perturbed or OOD test sets. I believe this will provide a more accurate and practical assessment of your method’s robustness.

---

> ### Author Response · Authors · 2025-08-03
>
> Dear Reviewer,
>
> We sincerely appreciate your patient response and valuable clarification. In response to your concerns, we provide the following detailed reply:
>
> **1. Adjusting Threshold Selection for Standard F1 Scores in Table 2**
>
> We agree with your comment that "the threshold chosen directly on the test set may lead to overly optimistic F1-scores." To ensure a fairer performance comparison, we select fixed thresholds for all methods based on scores computed on a separate clean dataset (e.g., DNA-DetectLLM: 0.6533, Binoculars: 0.9366, etc.). Subsequently, we recompute and report the F1 scores in Table 2 using these fixed thresholds.
>
> P.S. The clean dataset used for threshold selection consists of over 3,000 samples generated by GPT-4, Gemini, and Claude based on human-written texts sourced from XSum, WritingPrompt, and Arxiv. While this dataset is different from the data employed in the robustness experiments and ablation studies, it remains within the same distribution.
>
> Table 1-1 below presents the updated F1-scores from Table 2 after reselecting thresholds.
>
> Table 1-1
> | Method| M4| DetectRL Multi-LLM | DetectRL Multi-Domain | RealDet | Avg.   |
> |--|--|--|--|-|-|
> | ***OpenAI-D***  | 68.01  | 70.55 | 67.89| 70.41   | 69.22  |
> | ***Biscope***  | 71.75  | 72.00| 68.91| 81.23   | 73.97  |
> | **R-Detect*** | 67.14  | 66.56| 66.26 | 67.55   | 66.88  |
> | ***Entropy*** | 75.39  | 70.10| 55.04| 60.33   | 65.22  |
> | ***Likelihood***  | 66.82  | 66.62| 66.60| 66.87   | 66.73  |
> | ***LogRank*** | 66.80  | 66.69 | 66.60| 66.82   | 66.73  |
> | ***DetectGPT***  | 54.53  | 42.41| 30.79| 66.59   | 48.58  |
> | ***Fast-DetectGPT*** | 81.27  | 75.84| 68.06| 84.72   | 77.47  |
> | ***Binoculars***| 84.82  | 80.97 | 76.24| 82.15   | 81.05  |
> | ***Lastde++*** | 82.74  | 69.17| 61.72| 84.59   | 74.56  |
> | ***DNA-DetectLLM***| **85.15**   | **84.49**| **83.94**| **84.72**| **84.58**|
>
> **2. Additional Reporting of F1-scores in Robustness/Ablation Experiments**
>
> Using the fixed thresholds selected as described above, we additionally report the F1-score performances for the primary robustness experiments (as shown in Tables 1-2). Notably, DNA-DetectLLM consistently exhibits superior robustness compared to other baselines in practical scenarios. For a more accurate and practical evaluation, we will supplement these F1-score results along with other robustness and ablation experiment results in the appendix of our final manuscript.
>
> Interestingly, during our experiments, we observed that **methods with stronger performance typically possess more stable thresholds** (i.e., thresholds with smaller fluctuations). DNA-DetectLLM’s stable threshold further ensures its generalization capability across various data distributions.
>
> Once again, we sincerely thank you for your constructive suggestions. We genuinely hope our responses adequately address your concerns and look forward to further interactions with you.
>
> Table 1-2
> | Method | GPT-4 Insert | Deletion | Substitution | Paraphrase | Gemini Insert | Deletion | Substitution | Paraphrase | Claude Insert | Deletion | Substitution | Paraphrase | Avg.|
> |-|--|--|-|-|-|-|-|--|-|-|--|-|--|
> | ***OpenAI-D***| 57.17| 58.23  | 58.91   | 74.24  | 59.52| 60.24  | 60.34   | 72.99  | 65.00 | 66.04  | 65.15   | 71.93  | 64.15  |
> | ***Biscope***| 54.31 | 55.19  | 50.76   | 27.09  | 86.46| 86.34  | 86.30   | 42.42  | 86.09 | 85.80  | 85.80   | 32.73  | 64.94  |
> | ***R-Detect***| 66.46  | 66.46  | 66.46   | 66.50  | 66.16| 66.16  | 66.16   | 66.16  | 66.63 | 66.63  | 66.63   | 66.63  | 66.42  |
> | ***Entropy***| 62.53 | 65.04  | 61.92   | 56.82  | 65.95| 68.69  | 64.68   | 57.09  | 68.27| 71.36  | 67.02   | 57.20  | 63.88  |
> | ***Likelihood***| 66.65| 66.65  | 66.65   | 66.61  | 66.54| 66.54  | 66.54   | 66.54  | 66.69| 66.69  | 66.69   | 66.65  | 66.62  |
> | ***LogRank***| 66.61| 66.61  | 66.61   | 66.61  | 66.57| 66.57  | 66.57   | 66.50  | 66.65| 66.65  | 66.65   | 66.65  | 66.60  |
> | ***DetectGPT*** | 21.80| 30.87  | 17.13   | 21.04  | 54.86 | 65.84  | 52.22   | 30.18  | 43.03 | 54.86  | 38.79   | 22.13  | 37.73  |
> | ***Fast-DetectGPT***   | 77.10 | 84.28  | 76.24   | 89.38  | 87.90| 89.04  | 87.85   | 88.03  | 77.00 | 85.22  | 75.81   | 84.80  | 83.55  |
> | ***Binoculars***| 81.33| 87.16  | 81.15   | 91.50  | 92.44| 93.08  | 92.28   | 91.65  | 83.15 | 88.71  | 82.36   | 88.78  | 87.80|
> | ***Lastde++***| 60.86| 80.67  | 60.39   | 87.33  | 82.46 | 87.16  | 81.88   | 85.87  | 61.46| 82.22  | 61.26   | 83.02  | 76.22  |
> | ***DNA-DetectLLM***| **86.74** |**90.36** | **86.01**   | **93.09**  | **94.22**| **94.94**  | **93.91**   | **93.63**  | **87.28**| **91.58**  | **86.81**   | **91.06**  | **90.80**  |
>
> Best regards,
>
> All authors

---

> ### Comment · Reviewer_LFW4 · 2025-08-04
> **Official Comment by Reviewer LFW4**
>
> I appreciate the authors’ additional experiments and clarifications regarding my concerns, and I am glad to see that most of my issues with the paper have been addressed. I have therefore raised my score to 5.
>
> In fact, I understand the original experimental approach taken by the authors, which is consistent with many previous excellent works such as DetectGPT and Fast-DetectGPT. However, in practical testing, these detectors’ performance is far from sufficient for real-world applications. I think the main reason is that the evaluation metrics and test settings make the experimental results appear overly optimistic. From the updated results provided by the authors, I believe this work already offers enough insight and justification for acceptance, including its interesting motivation, encouraging experimental results and robustness, as well as manageable computational overhead. Additionally, I very much appreciate the authors’ experimental design, especially the validation of the detector’s robustness on more challenging benchmarks such as DetectRL, which is highly instructive.
>
> I hope the authors can adjust the experimental setup and update all experimental results according to the rebuttal (if this paper is accepted), as this will be important for readers who wish to follow up on your research. As I mentioned, I appreciate the experimental design of this paper and hope it can serve as a standard for future research.
>
> Finally, the only slight shortcoming in the paper is that the performance of the supervised detectors is very poor, which seems unreasonable because the comparison is unfair. As you mentioned, all the statistical-based methods are trained on in-distribution datasets, but the supervised detectors are not. In fact, if these detectors were trained on data from the same distribution, their performance would be very good, even with only a small amount of data, in my experience. That said, I think the authors should clearly state in the paper that the settings for these supervised detectors are different from those for the statistical methods. This will help readers understand the out-of-distribution detection limitations of such detectors, rather than giving the impression of downplaying the performance of supervised detectors to highlight the advantages of statistical methods. Otherwise, this may mislead readers into thinking that supervised detectors are very poor, though in reality their practical applicability may be much higher than that of statistical methods, especially on in-distribution data.

---

> > ### Author Response · Authors · 2025-08-05
> >
> > Dear Reviewer:
> >
> > We are very pleased to see your positive evaluation of our work, and we sincerely thank you once again for the time and effort you have devoted to improving this manuscript. We will incorporate the above adjustments faithfully into the final version.
> >
> > Best regards,
> >
> > All authors

---

### Official Review · Reviewer_284H · 2025-07-03

**Clarity:** 3
**Significance:** 3
**Originality:** 2
**Rating:** 5
**Confidence:** 4

**Summary:**

The paper proposes a new method for the unsupervised detection of LLM-generated text. The proposed method is a modified version of the cross-perplexity method from Binoculars [1]; however, instead of computing the ratio of perplexity to cross-perplexity, it computes the "conditional score", i.e., the ratio of the probability of argmax tokens given the observed document with the cross-perplexity.

[1] Hans, A., Schwarzschild, A., Cherepanova, V., Kazemi, H., Saha, A., Goldblum, M., ... & Goldstein, T. (2024, July). Spotting LLMs with binoculars: zero-shot detection of machine-generated text. In Proceedings of the 41st International Conference on Machine Learning (pp. 17519-17537).

**Questions:**

N/A

**Ethical Concerns:**

["NO or VERY MINOR ethics concerns only"]

**Final Justification:**

Strong results, very thorough experiments, and good generalization between disparate scoring and target models.

**Limitations:**

yes

**Quality:**

3

**Strengths And Weaknesses:**

Strengths:

1. The paper is generally well-written and (mostly) easy to understand. It does a good job of differentiating between unsupervised and supervised baselines and reports competitive approaches in both categories

2. The results are really quite impressive given the large difference between the scoring models (Falcon-7B, LLaMA-7B) and the target models (GPT-4-turbo, Gemini-2.0-flash, Claude-3.7-Sonnet). This is in pretty stark contrast to some existing methods like DetectGPT, which experience large performance degradations when the scoring and target models don't match

3. The paper experiments on a wide range of benchmarks, models, and baseline approaches. It includes experiments on paraphrasing and editing attacks, analyzes performance as a function of the number of tokens, and evaluates different scoring methods. Very thorough!

Weaknesses:

1. I found the "DNA" inspiration to be more confusing than helpful. As far as I can tell, the "ideal sequence" here is just a sequence of argmax tokens which are selected independently of one another, so the ideal sequence might actually be very low probability or ungrammatical under a language model

2. The method in this paper seems to be pretty closely motivated by Binoculars. While I think this is fine, it could be more explicit about it and justify some of the decisions made in the Binoculars paper (e.g., why cross-perplexity) in order to be more easily read as a standalone paper. For example, my understanding is that the models experimented on are paired as (Falcon-7B-Instruct, Falcon-7B) and (Llama-2-7B, Llama-7B) because the cross-perplexity computation requires models to share a tokenizer, but this is not explained in the paper

---

> ### Author Rebuttal · Authors · 2025-07-26
>
> Dear Reviewer,
>
> We sincerely appreciate the time and effort you have dedicated to reviewing our manuscript and providing valuable suggestions. In response to your feedback, we offer the following point-by-point replies:
>
> **R1 (to W1):** The ideal sequence is derived from the input text and corresponds one-to-one with it. When the ideal sequence exists independently as a text sequence, it may have a lower probability or deviate from grammatical rules due to independent selection. However, from a fine-grained perspective, each token in the ideal sequence represents the most likely token for the corresponding position in the input text, similar to how nucleotides on the template strand of DNA replication are healthy (non-mutated). In contrast, the tokens (those not of the maximum probability) in the original text may correspond to mutated nucleotides on the DNA replication strand. These inspired our thinking to introduce the mutation-repair paradigm, analogous to DNA, into the detection process.
>
> **R2 (to W2):** Expand on the Binoculars score to further clarify and strengthen the paper’s readability. The mutation score in DNA-DetectLLM improves upon the Binoculars score because the latter (the ratio of log perplexity to cross-perplexity) is highly effective at capturing AI-generated text with high perplexity. A key condition for calculating the cross-perplexity from dual model perspectives is that the tokenizers for the LLMs must be identical. We will provide a detailed explanation of the above points in Section 3.1 (Preliminary) of the final version of the manuscript.
>
> Best regards,
>
> All authors

---

> > ### Comment · Reviewer_284H · 2025-08-05
> >
> > Thanks for the clarifications! I will keep my (already high) score.

---

### Official Review · Reviewer_gfB8 · 2025-07-04

**Clarity:** 3
**Significance:** 3
**Originality:** 3
**Rating:** 5
**Confidence:** 4

**Summary:**

The propose approach is a zero shot approach for LLM generated sequence detection. Given an input sequence (text utterance), ideal AI generated sequence is obtained using greedily selecting the most probable token at each position using probability distribution obtained using Falcon-7B-Instruct. Once both ideal AI generated sequence and original input sequence is available. The position where there is a change is compared and referred as mutations. The input sequence is aligned with ideal LLM generated sequence using an iterative token-level modifications referred as mutation repair mechanism. A score is accumulated throughout the repair. The accumulated score is compared against a calibrated threshold to label input sequence as AI generated or human written. Author run exhaustive experiments on multiple dataset and compare with several existing work to show competitive performance from proposed approach.

**Questions:**

Refer above

**Ethical Concerns:**

["NO or VERY MINOR ethics concerns only"]

**Final Justification:**

I thank author for clearing minor concerns about LLM pairing I had and concerns from other reviewers. I look forward to reading more details about it in Appendix in the final version. I stay with my initial recommendation to accept.

**Limitations:**

Yes

**Quality:**

3

**Strengths And Weaknesses:**

Major Strength:

1. Approach is novel, there are existing work on DNA inspired zero shot algorithms for AI generated text detection e.g. R1 and R2. However, proposed approach is different w.r.t them.
2. Performance from proposed approach is competitive and author compare with several recent methods also like Lastdee++. Author also run experiments to show method is more robust than existing method from attacks like insertion, deletion, substitution and paraphrase.
3. Since method is based on accumulated score between ideal AI sequence and input sequence. Existing work show that detection performance can be sensitive to length of the sequence. Detection become more challenging for smaller length sequence. Author show proposed approach is robust to different sequence lengths
4.  Author also show that proposed approach works independent of base LLM used to generate ideal AI generated sequence.
5. Paper is well written and easy to understand.

Minor Weakness: Using different base LLMs and reporting detection performance using them in Table 2 or Appendix can be helpful.

R1: https://arxiv.org/html/2305.12519v2
R2: https://openreview.net/forum?id=Xlayxj2fWp

---

> ### Author Rebuttal · Authors · 2025-07-26
>
> Dear Reviewer,
>
> We sincerely appreciate the time and effort you have dedicated to reviewing our manuscript and providing valuable suggestions. We are truly delighted by your positive assessment of our work. In response to your feedback, please find our reply below:
>
> **Response to Weakness:** Figure 6 in our manuscript illustrates the performance of DNA-DetectLLM using various base LLMs on the DetectRL dataset. We have identified that further improvements can be achieved through optimal LLM pairing. To better highlight and clarify these experimental findings, we will include an additional comparative table in the Appendix in the final version.
>
> Best regards,
>
> All authors

---

### Note · Authors · 2025-08-12

Dear PC, SAC, AC, and Reviewers,

We sincerely thank the AC and all reviewers for the time and effort you have devoted to evaluating our paper. Following the in-depth discussions during the rebuttal phase, we have further strengthened our work in response to the reviewers’ concerns. Below, we summarize the key consensus strengths identified in the reviews, as well as the updates and improvements made during the rebuttal:

**Consensus Strengths from the Reviews**

**1. Well-motivated and Novel Method:**  We propose a DNA-inspired perspective, leveraging a repair-based process to directly and interpretably capture the intrinsic differences between human-written and AI-generated text.

**2. The Paper is well written and easy to understand:** All reviewers agreed that the paper is easy to follow and have a high level of confidence.

**3. Comprehensive Experimental Evaluation:** Our work compares against a broad range of baseline methods, challenging public datasets, and diverse detection scenarios, while also including necessary ablation studies.

**4. Competitive Experimental Results:** The results demonstrate that DNA-DetectLLM achieves state-of-the-art detection performance, showing greater robustness under different attacks and text lengths, while maintaining low time cost. Overall, it stands as a highly competitive zero-shot detection method.

**Updates and Improvements During the Rebuttal Phase**

**1. Minior F1-Score Adjustment and Supplement:** We selected a unified decision threshold to calculate F1-score in Table 2 based on the method’s performance on the clean dataset and incorporated additional F1-score evaluations in the robustness experiments.

**2. Additional Performance Comparisons:** Following reviewers’ suggestions, we added comparisons with Revise-Detect, GECScore, DNA-GPT, IMBD (recent but non-SOTA baselines), and GPTZero (a commercial detection tool), and included results on the biomedical short-text dataset PubMedQA, further highlighting the strong performance of DNA-DetectLLM.

**3. Additional Details for Reproducibility and Readability:** We included explicit declarations of API parameters to enhance reproducibility, and added a more detailed introduction to the Binoculars score to improve readability.

We once again sincerely thank the reviewers for their positive discussions and unanimous recognition during the rebuttal phase.

Best regards,

All authors

---

### Decision · Program_Chairs · 2025-09-17

**Decision:**

Accept (spotlight)

**Comment:**

This paper introduces a clever new way to detect AI-generated text. Inspired by DNA repair, the method, called DNA-DetectLLM, measures the effort needed to fix a piece of text to make it look like an LLM-generated sequence. The core idea is that human writing, with its natural imperfections and stylistic choices, will require a lot more "repair" than text that was already generated by an LLM.

The reviewers appreciated the idea and the impressive results. In response to feedback from the reviewers, the authors added more baselines, including recent academic work like DNA-GPT and ImBD, and importantly, the commercial tool GPTZero. They also tested their method on the PubMedQA dataset, showing robustness in a completely different domain (short-form biomedical text). They also adopted a more rigorous fixed-threshold method for calculating F1 scores across all experiments,

I think this is a strong submission and appreciate how this approach doesn't require any training and is interpretable as well.